# Efficacy and Safety of Antibiotics in the Treatment of Methicillin-Resistant *Staphylococcus aureus* (MRSA) Infections: A Systematic Review and Network Meta-Analysis

**DOI:** 10.3390/antibiotics13090866

**Published:** 2024-09-10

**Authors:** Qi Liu, Dongxia He, Lei Wang, Yuewei Wu, Xian Liu, Yahan Yang, Zhizhi Chen, Zhan Dong, Ying Luo, Yuzhu Song

**Affiliations:** 1College of Life Science and Technology, Kunming University of Science and Technology, Kunming 650500, China; liuqi7@stu.kust.edu.cn (Q.L.); hedongxia@stu.kust.edu.cn (D.H.); 20222118052@stu.kust.edu.cn (L.W.); wuyuewei@stu.kust.edu.cn (Y.W.); liuxian@stu.kust.edu.cn (X.L.); yangyahan@stu.kust.edu.cn (Y.Y.); chenzhizhi@stu.kust.edu.cn (Z.C.); 20200040@kmmu.edu.cn (Z.D.); luoying@stu.kust.edu.cn (Y.L.); 2Graduate School, Kunming University of Science and Technology, Kunming 650500, China

**Keywords:** antibiotics, MRSA infection, vancomycin, treatment, network meta-analysis

## Abstract

Background: Vancomycin is a first-line drug for the treatment of MRSA infection. However, overuse of vancomycin can cause bacteria to become resistant, forming resistant strains and making infections more difficult to treat. This study aimed to evaluate the efficacy and safety of different antibiotics in the treatment of MRSA infections and to compare them, mainly with vancomycin, to find better vancomycin alternatives. Methods: All studies were obtained from the PubMed and Embase databases from inception to 13 April 2023. The three comprehensive indicators of clinical cure success rate, clinical microbiological success rate, and adverse reactions were evaluated, and the clinical cure success rates of three disease types, complex skin and skin structure infections (cSSSIs), complex skin and soft tissue infections (cSSTIs), and pneumonia, were analyzed in subgroups. All statistical analyses were performed using R and STATA 14.0 software for network meta-analysis. Results: A total of 38 trials with 6281 patients were included, and 13 drug treatments were evaluated. For MRSA infections, the results of network meta-analysis showed that the clinical success rates of linezolid, the combination of vancomycin and rifampin, and the combination of minocycline and rifampin were better than that of vancomycin (RR 1.71; 95%-CI 1.45–2.02), (RR 2.46; 95%-CI 1.10–5.49) (RR, 2.77; 95%-CI 1.06–7.21). The success rate of clinical microbiological treatment with vancomycin was inferior to that with telavancin (RR 0.74; 95%-CI 0.55–0.99). Linezolid had a higher rate of adverse reactions than teicoplanin (RR 5.35; 95%-CI 1.10–25.98). Subgroup analysis showed that vancomycin had a lower clinical success rate than linezolid in the treatment of MRSA-induced cSSSIs, cSSTIs, and pneumonia (RR 0.59; 95%-CI 0.44–0.80) (RR 0.55; 95%-CI 0.35–0.89) (RR 0.55; 95%-CI 0.32–0.93). Conclusions: This systematic review and NMA provide a new comparison framework for the clinical treatment of MRSA infection. The NMA suggests that linezolid may be the antibiotic of choice for the treatment of MRSA infections, with the ability to improve clinical and microbiological success rates despite its disadvantage in terms of adverse effects. At the same time, the combination of minocycline and rifampicin may be the most effective drug to treat MRSA-induced cSSSIs, tedizolid may be the best drug to treat MRSA-induced cSSTIs, and the combination of vancomycin and rifampicin may be the most effective treatment for MRSA-induced pneumonia. More high-quality studies are still needed in the future to further identify alternatives to vancomycin. Trial registration: PROSPERO registration number CRD42023416788.

## 1. Background

Infection caused by methicillin-resistant *Staphylococcus aureus* (MRSA) was first identified in 1961 [1]. For nearly 60 years, because of the overuse of antibiotics, rates of hospital- and community-acquired infections caused by MRSA have risen [2,3]. At present, the main reasons for which MRSA infection has become a global problem include complex skin and skin structure infections (cSSSIs), complex skin and soft tissue infections (cSSTIs), pneumonia, and bacteremia. A study in the United States found that MRSA was isolated from nearly 60% of the patients with cSSSIs, which are among the most common treatment-associated infections in the medical system, with high morbidity and healthcare costs [4,5]. At the same time, MRSA infection accounts for 60% of cSSTIs, which are one of the fastest-growing causes of hospitalization and impose a large economic burden [6,7]. With this series of problems that threaten the lives and health of patients, medical institutions are in urgent need of more effective treatment programs to solve these problems.

Vancomycin is the most commonly used antibiotic in the treatment of MRSA infection, but the overuse of vancomycin has led to the emergence of resistant strains, influencing the treatment’s efficacy [8]. Currently, available antibiotics include rifampin, doxycycline, minocycline, ceftaroline, clindamycin, teicoplanin, TMP-SMX, vancomycin, daptomycin, tigecycline, quapritin/dalfopritin, and linezolid [9,10,11]. The European and American guidelines recommend vancomycin and linezolid as first-line treatments [12,13]. The curative effect of this class of antibiotics is mostly for patients with suspected or confirmed MRSA infection, as well as patients not diagnosed with MRSA infection.

In this study, we conducted a network meta-analysis (NMA) to compare the efficacy and safety of different antibiotics against vancomycin in the treatment of MRSA infection in order to determine which antibiotic is the best alternative to vancomycin.

## 2. Objective

The main objective of this review was to evaluate the efficacy and safety of clinical antibiotic drugs in patients with MRSA infection. In this paper, we comprehensively evaluate the efficacy and safety of antibiotic drugs in the clinical treatment of MRSA-infected patients. This study aims to guide the current clinical use of antibiotics for the treatment of MRSA infections.

## 3. Results

### 3.1. Study Selection

We searched the PubMed and Embase databases for a total of 18,398 potentially relevant articles. After excluding 1913 duplicate articles, the full-text contents of 358 articles were reviewed after reading the titles and abstracts. These articles were evaluated in full, and finally, we included 38 eligible studies that met the criteria (see Figure 1).

### 3.2. Study Characteristics

The main characteristics of the included studies are shown in Table 1, with 38 articles including a total of 6281 patients, and all study participants were patients with confirmed MRSA infection. For treatment of the related diseases caused by MRSA infection, a total of 17 antibiotics were evaluated as monotherapy or combination therapy.

### 3.3. Bias Risk Assessment

Of the 38 articles included, 37 were clear RCT research studies. One of the studies was not an RCT and was not double-blinded, posing a high risk of bias. Overall, the included articles had a low-to-moderate risk of bias (see Appendix A).

### 3.4. The Clinical Cure Rate

Of the 38 included articles, a total of 29 articles evaluated the clinical cure success of 4097 patients, 2825 of whom were treated successfully (Table 2). The direct meta-analysis results are shown in Appendix A. To further determine the direct and indirect comparative efficacy of these 13 antibiotics, we performed a network meta-analysis; the network evidence graph is shown in Figure 2A. The results showed that the clinical success rate of linezolid was better than that of vancomycin (RR 1.71; 95%-CI 1.45–2.02). The clinical success rate of vancomycin combined with rifampin was better than that of vancomycin alone (RR 2.46; 95%-CI 1.10–5.49). The clinical success rate of minocycline combined with rifampin was better than that of vancomycin (RR 2.77; 95%-CI 1.06 7.21), and the other antibiotics showed no statistical significance (see Figure 3 and Figure 4A).

We sorted the antibiotics according to their probability of success in clinical cure. The results showed that the combination of minocycline and rifampin (80.4%) had the highest probability of being first, followed by the combination therapy of vancomycin and rifampin (77.2%), tedizolid (70.3%), telavancin (66.4%), linezolid (64.3%), teicoplanin (50.7%), cefadroxil (45.8%), TMP-SMZ (44.7%), vancomycin (40.1%), daptomycin (34. 7%), arbekacin (26.1%), ceftaroline (25.5%), and vancomycin and aztreonam (23.8%). Rank probability plots and rank cumulative probability plots for all antibiotics are shown in Figure 5A and Appendix A.

### 3.5. Clinical Microbiology Success Rate

Of the 38 articles included, a total of nine articles, including 2289 cases, evaluated the clinical microbiological success, with 2001 cases of successful treatment (Table 2). The direct meta-analysis results are shown in Appendix A. To further compare the direct and indirect effects of five kinds of antibiotics, we conducted a network meta-analysis, and the evidence network diagram is shown in Figure 2B. The results showed that the success rate of clinical microbiological treatment with vancomycin was inferior to that with telavancin (RR 0.74; 95%-CI 0.55–0.99), which is consistent with the results of the direct meta-analysis, and no statistical significance was found for the remaining antibiotic comparisons (see Figure 4B and Appendix A).

We sorted the antibiotics according to their probability of success in clinical microbiology. The results showed that linezolid (78.8%) was the most likely to be ranked first, followed by telavancin (61.6%), ceftobiprole (50.1%), tigecycline (34.7%), and vancomycin (24.8%). Rank probability plots and rank cumulative probability plots for all antibiotics are shown in Figure 5B and Appendix A.

### 3.6. Incidence of Adverse Reactions

Among the 38 included articles, 10 articles evaluated the adverse reactions of 1536 patients, finding that they occurred in 1051 patients (Table 2). The results of the direct meta-analysis are shown in Appendix A. To further compare the direct and indirect efficacy of these six antibiotics, we performed a network meta-analysis; the network evidence graph is shown in Figure 2C. The evidence figure directly or indirectly compares the six kinds of antibiotic monotherapy or combination therapy. The results show that linezolid had a higher rate of adverse reactions than teicoplanin (RR 5.35; 95%-CI 1.10–25.98), and no statistical significance was found for the remaining antibiotics (see Figure 4C and Appendix A).

We ranked the antibiotics according to their probability of adverse effects. The results showed that linezolid (77.9%) had the highest probability, followed by cefadroxil (71.9%), minocycline combined with rifampin (52%), vancomycin combined with rifampin (46.6%), vancomycin (44.4%), and teicoplanin (7.1%). The rank probability diagram and the rank cumulative probability diagram for all antibiotics are shown in Figure 5C and Appendix A.

### 3.7. Clinical Cure Rate of Patients with MRSA-Induced cSSSIs

Of the 38 included articles, 10 reviewed the clinical cure rate of 1479 patients with MRSA-induced cSSSIs, of whom 1085 were successfully cured (Table 2). The results of the direct meta-analysis are shown in Appendix A. To further compare the direct and indirect efficacy of these seven antibiotics, we conducted a network meta-analysis, and the network evidence graph is shown in Figure 2D. The evidence map directly or indirectly compares monotherapy or combination therapy with seven antibiotics; the results show that the clinical success rate of vancomycin was worse than that of linezolid (RR 0.59; 95%-CI 0.44–0.80), while the clinical success rate of combined minocycline and rifampin treatment was superior to that of vancomycin (RR 2.74; 95%-CI 1.02–7.38), and no statistical significance was found for the remaining antibiotic comparisons (see Figure 6A and Appendix A).

We ranked the antibiotics according to their probability of clinical cure. The results showed that the combination treatment with minocycline and rifampin (90.6%) ranked first, followed by linezolid (76.5%), vancomycin (48.6%), daptomycin (41%), cefadroxil (38.4%), ceftaroline (28.5%), and vancomycin and aztreonam (26.3%). The rank probability diagram and the rank cumulative probability diagram for all antibiotics are shown in Figure 5D and Appendix A.

### 3.8. Clinical Cure Rate of Patients with MRSA-Induced cSSTIs

Among the 38 included articles, a total of eight articles evaluated the clinical cure rate of 1397 patients with MRSA-induced cSSTIs, of whom 1055 were successfully cured (Table 2). The results of the direct meta-analysis are shown in Appendix A. To further compare the direct and indirect efficacy of these five antibiotics, we conducted a network meta-analysis, and the network evidence graph is shown in Figure 2E. The evidence map directly or indirectly compares the effects of monotherapy with five antibiotics; the results show that the clinical success rate of vancomycin was worse than that of linezolid (RR 0.55; 95%-CI 0.35–0.89), consistent with the results of the direct meta-analysis, while the remaining antibiotic comparisons were not found to be statistically significant (see Figure 6B and Appendix A).

We ranked the antibiotics according to their probability of clinical cure. The results showed that tedizolid (70.7%) was the most likely successful treatment, followed by linezolid (66.8%), telavancin (64.8%), vancomycin (29.7%), and arbekacin (18%). The rank probability diagram and the rank cumulative probability diagram for all antibiotics are shown in Figure 5E and Appendix A.

### 3.9. Clinical Cure Rate of MRSA-Induced Pneumonia Patients

Of the 38 included articles, seven evaluated the clinical cure rate in 955 patients with MRSA-induced pneumonia, of whom 449 were clinically cured (Table 2). The results of the direct meta-analysis are shown in Appendix A. To further compare the direct and indirect efficacy of these four antibiotics, we conducted a network meta-analysis, and the network evidence graph is shown in Figure 2F. The evidence map directly or indirectly compares the effects of the four antibiotics administered alone or in combination; the results show that the clinical success rate of vancomycin was worse than that of linezolid (RR 0.55; 95%-CI 0.32–0.93), consistent with the results of the direct meta-analysis, while the remaining antibiotic comparisons were not found to be statistically significant (see Figure 6C and Appendix A).

We ranked the antibiotics according to their probability of clinical cure. The results showed that the combination of vancomycin and rifampin (68.8%) ranked first, followed by cefadroxil (62.7%), linezolid (56.2%), and vancomycin (12.2%). The rank probability diagram and the rank cumulative probability diagram for all antibiotics are shown in Figure 5F and Appendix A.

### 3.10. Publication Bias and Inconsistency Evaluations

We observed the symmetry of the funnel plot to detect publication bias, finding that it had a generally symmetric distribution, and no significant publication bias was found in any of the results, while there were three points distributed outside the 95%-CI, indicating the possible influence of the small sample size (see Appendix A). Heterogeneity or inconsistencies could not be assessed because no closed loops were present in any of the study networks.

## 4. Methods

### 4.1. Search Strategy

We systematically searched PubMed and EMBASE for potentially eligible studies (the database was established until April 2023) using Medical Subject Headings (MeSH) descriptors combined with free-text terms to retrieve two categories: MRSA and treatment drugs. In addition, the references of all included studies were manually searched, and a meta-analysis was performed on the same topic to identify additional eligible studies (see Appendix A in the Appendix A).

### 4.2. Choice Criteria

This study included articles that complied with the following criteria: (1) RCTs or observational studies published in English. (2) Studies including patients diagnosed with MRSA infection (the diagnosis was confirmed by Gram staining, culture, and drug-sensitivity test), as well as patients with cSSSIs, cSSTIs, and pneumonia caused by MRSA infection, comparing antibiotics with anti-MRSA activity against one another. (3) The primary outcomes were the rate of clinical and microbiologic cure at the test-of-cure visit and the rate of adverse effects. In addition, we excluded relevant literature in which only antibiotic experiments were conducted.

### 4.3. Data Extraction

For each included study, data extraction was performed by two independent reviewers using a standardized data-extraction form, and the relevant study characteristics extracted included (1) the first author, (2) publication year, (3) research types, (4) disease types caused by MRSA infection, (5) the basic information of the patients, (6) medication information (i.e., medication method, dosage, and duration), (7) prognosis and efficacy (i.e., clinical and microbiological cure rates), and (8) adverse reactions.

### 4.4. Outcome Indicators

We considered clinical success, microbiological success, and adverse event (AE) outcomes in the network meta-analysis. First, the populations tested for clinical success included the intention-to-treat (ITT) population, the modified intention-to-treat (MITT) population, the clinically evaluable (CE) population, and some populations that were not specified in the study. The ITT population is defined as those who received at least one dose of the study drug, while all randomized and MITT patients were defined at baseline to confirm the presence of MRSA in all ITT patients. Clinical success was defined as cure or improvement to assess the status of the study population upon the test of cure (TOC), while TOC is assessed 7–14 days after the end of treatment. For microbiological success, we assessed the microbiologically evaluable (ME) population as patients who had at least one MRSA pathogen isolated from blood or infected tissue at baseline in the CE population, with microbiological success defined as the eradication of MRSA pathogens. Finally, safety, defined as the incidence of adverse events, was assessed. Some of the common adverse reactions include nausea, diarrhea, headache, and thrombocytopenia.

### 4.5. Quality Evaluation

The quality of the research was evaluated by two independent reviewers according to the Cochrane Handbook assessment tools; when necessary, disputes between the two independent reviewers were resolved through discussion with the third reviewer. Among the items evaluated in the manual were seven areas: random sequence generation, allocation concealment, blinding of participants and personnel, blinding of outcome assessment, incomplete outcome data, selective reporting, and other biases. Each indicator was evaluated separately using low, unclear, or high risk of bias. Finally, Review Manager 5.4 software was used to generate the risk-of-bias map.

### 4.6. Statistical Analysis

We evaluated three outcome measures, as well as subgroup analyses of three diseases in clinical success measures through a network meta-analysis, comparing the effectiveness and safety of various antibiotics for the treatment of MRSA infections. A meta-analysis of bicategorical variables was performed using the meta software package in R language. Relative risks (RRs) and 95% confidence intervals (95%-CIs) were calculated to compare the efficacy and safety of various antibiotics in a specified population. If 95%-CI of RR does not contain 1, it is considered that there is a statistically significant difference. The size of the heterogeneity was assessed using I^2^. If I^2^ < 50%, the effect size was combined using a fixed effect model, and if I^2^ ≥ 50%, the effect size was combined using a random effect model. Using Stata 14.0 software, a network meta-analysis was conducted based on the random effects model under the frequency framework, and the network evidence map was drawn. Subsequently, the cumulative ranking probability area under graph (SUCRA) was calculated by the “sucra prob” command to predict the efficacy ranking of each intervention. The larger the SUCRA value, the better the effectiveness of the intervention so as to facilitate the direct and indirect comparison of the efficacy of various antibiotics. Finally, the “netfunnel” command was used to plot funnel plots to evaluate publication bias in the included literature. This study’s PROSPERO registration number is CRD42023416788.

## 5. Discussion

This systematic review and the NMA evaluated the efficacy and safety of different antibiotics for the treatment of infections caused by MRSA. We classified the 38 included studies, and the NMA results showed that the clinical cure success rates of minocycline combined with rifampin, vancomycin combined with rifampin, and linezolid were superior to that of vancomycin. The SUCRA results indicated that combined treatment with minocycline and rifampin may have the highest clinical cure success rate. The efficacy of linezolid was higher than that of vancomycin in the evaluation of clinical microbial success rate and the incidence of adverse reactions. In addition, the results of the subgroup analysis showed that the clinical cure rates of linezolid combined with minocycline and rifampin were better than that of vancomycin in the treatment of cSSSI patients, and the clinical cure rate of linezolid in the treatment of cSSTI patients and pneumonia patients was better than that of vancomycin.

Three similar studies were conducted previously. In 2012, Bally et al. compared the efficacy of six antibiotics used to treat cSSTIs and HAP/VAP (hospital-acquired pneumonia/ventilator-associated pneumonia), finding that linezolid had the best efficacy against cSSTIs, while the efficacy of linezolid was better than that of vancomycin in treating pneumonia, consistent with the results of this study [52]. The Bayesian model emphasizes the introduction of subjective prior information, so different people may have had different prior distributions, resulting in different inferred results. In 2021, Feng et al. suggested that linezolid may be the antibiotic of choice for the treatment of skin and soft tissue infections (SSTIs) caused by MRSA, which is consistent with the findings of this study [53]. However, only 8 of the 20 trials included were blinded, and 16 of them were conducted in the United States, potentially limiting the results. In 2024, Ju et al. compared the efficacy of six antibiotics in the treatment of MRSA infections and found that linezolid was the most effective in the treatment of lung infections, as well as skin and soft tissue infections, which is generally consistent with the results of this study [54]. Compared with the previous three studies, the present study focused on the diagnosis of MRSA infections and conducted a reasonable classification analysis for different types of MRSA infections. We analyzed 13 antimicrobial agents, whether clinically used or not, to provide stronger and more comprehensive evidence for clinical use.

We found that the combination of minocycline and rifampicin in the treatment of MRSA infections had the best clinical cure rate, followed by linezolid, both of which were superior to vancomycin. The combination treatment with minocycline and rifampicin is mainly used due to the good absorption of these two drugs, along with their strong tissue permeability, long half life, and synergistic effect in fighting MRSA infection [55]. The incidence of adverse reactions is also relatively low, and the most common adverse reactions include metabolic and nutritional disorders [51]. The 2011 Infectious Diseases Society of America (IDSA) guidelines do not recommend the use of rifampicin alone for the treatment of MRSA-infection-related diseases [56]. Therefore, the combination of minocycline and rifampicin may be a better choice for the treatment of MRSA infections.

In the study of patients with MRSA-induced cSSSIs, we found that the combination of minocycline and rifampicin had the best effect and a low incidence of adverse reactions, but there have been few clinical trials of this combination, and further research is needed. In addition, in patients with MRSA-induced cSSTIs, we found that linezolid was more effective than vancomycin, which is consistent with previous results [52,57]. However, the SUCRA results of this study indicate that tedizolid may be the antibiotic with the highest success rate in the treatment of cSSTIs. Tedizolid, a new oxazolidone antibiotic, is no less effective than linezolid and has a lower incidence of drug-related adverse reactions [58,59]. At present, there have been few studies on the use of tedizolid in the treatment of cSSTIs, and the discovery of SUCRA may increase the research on tedizolid in the treatment of cSSTIs in the future. Finally, for patients with MRSA-induced pneumonia, we found a significant increase in the efficacy of linezolid, consistent with the results of previously published meta-analyses [60]. However, our SUCRA results showed that the combination of vancomycin and rifampicin was the most effective and had a lower incidence of adverse reactions. The 2011 IDSA guidelines recommend intravenous vancomycin, linezolid, or clindamycin for the treatment of MRSA-induced pneumonia [56]. The updated guidelines for 2021 recommend considering rifampicin as an adjunct in the treatment of MRSA-induced pneumonia [61]. In their study, Diekema et al. found no significant change in rifampicin resistance rates over 11 years [62]. Therefore, rifampicin may be a potential adjuvant treatment for MRSA infection in the future.

MRSA has become one of the major multidrug-resistant bacterial pathogens causing cSSSIs and hospital-acquired infections, especially associated pneumonia. This study found that the best treatment drugs for different types of diseases caused by MRSA infection are also different. cSSSIs are skin infections involving deep soft tissue, mainly including surgical sites, traumatic wound infections, and widespread cellulitis. Vancomycin, the drug of choice against MRSA, has poor penetration and struggles to work effectively in soft tissue and bone [63]. In addition, cSSSIs are not solely related to MRSA. Hospital-acquired skin infections are generally associated with Staphylococcus aureus, Gram-negative bacteria, and anaerobic bacteria, with diabetic foot infections typically involving three to five bacterial species [64]. cSSTIs are diseases caused by bacterial invasion of the epidermis, dermis, or subcutaneous tissue and can present with a variety of clinical symptoms. They are mostly caused by a single microbial infection caused by Gram-positive or -negative bacteria, and they often occur at sites of damaged tissue blood perfusion, which is conducive to the growth and reproduction of pathogenic bacteria [65]. MRSA is an important causative pathogen of pneumonia. Pneumonia is a respiratory infection, and its clinical symptoms depend on the continuous concentration of antibiotics at the infection site. Therefore, antibiotics need to achieve a certain level of biological activity in the relevant parts of the lung and not be inactivated by pulmonary surfactants. The three different types of MRSA infection diseases included in this study have different infection sites in the body, and the pathogens are also different, so different drug treatments can exert significant effects.

There are some limitations to this study. First, this study was limited to English-language literature, and some non-English literature may have been missed. Second, there were some differences in the drug-treatment methods included in this study. Thirdly, the population included was diverse in terms of clinical cure success rates, including EOS, ITT, MITT, CE, and those that stated only clinical cure rates. In addition, there are currently insufficient data to demonstrate significant differences in the efficacy of other antibiotics in treating MRSA infections, and more high-quality clinical studies are needed to support these antibiotics as alternatives to vancomycin.

## 6. Conclusions

This systematic review and NMA provide a new comparative framework for the clinical treatment of MRSA infections. The NMA suggests that linezolid may be the antibiotic of choice for the treatment of MRSA infections, and although it has disadvantages in terms of adverse effects, it can improve clinical and microbiological success rates. At the same time, combination treatment with minocycline and rifampicin can also improve the clinical cure success rate and clinical cure rate of cSSSI patients. However, the comparative results of other antibiotics were not statistically significant, so the search for an alternative to vancomycin—the first-line drug for the treatment of MRSA infections—remains urgent, and more high-quality clinical studies are still needed in the future.

## Figures and Tables

**Figure 1 antibiotics-13-00866-f001:**
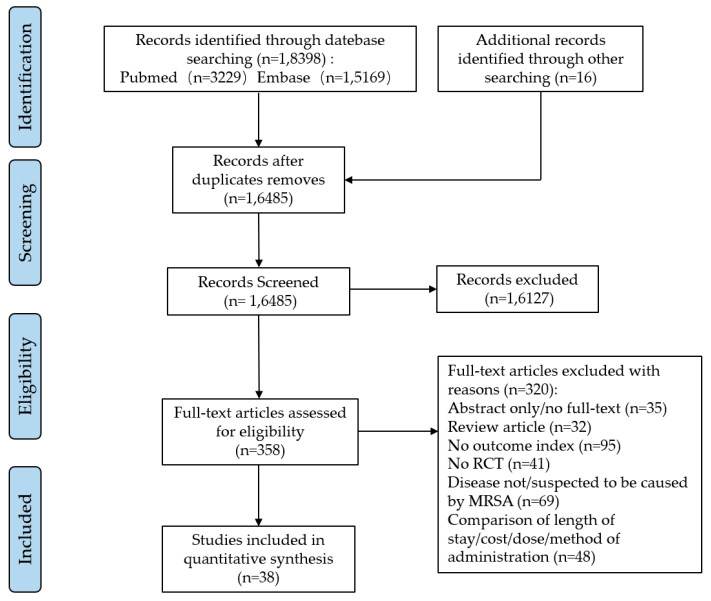
The flowchart of literature filtering.

**Figure 2 antibiotics-13-00866-f002:**
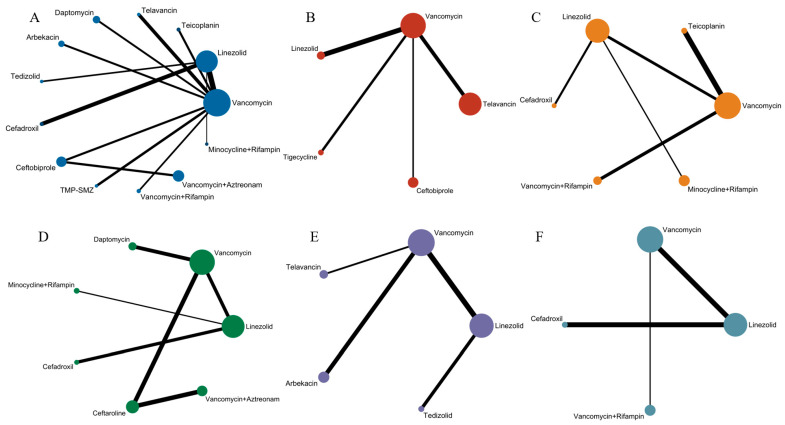
Network evidence plot, where the size of nodes corresponds to the cumulative sample size for individual antibiotics, and the thickness of the lines is proportional to the number of studies for each treatment comparison: (**A**) Evidence chart of clinical cure rate of 13 treatment methods: minocycline and rifampin, vancomycin and rifampin, tedizolid, telavancin, linezolid, teicoplanin, cefadroxil, TMP-SMZ, vancomycin, daptomycin, arbekacin, ceftaroline, and vancomycin and aztreonam. (**B**) Evidence chart of clinical microbiological success rate of five treatment methods: linezolid, telavancin, ceftobiprole, tigecycline, and vancomycin. (**C**) Evidence chart of incidence of adverse effects of six treatment methods: linezolid, cefadroxil, minocycline and rifampin, vancomycin and rifampin, vancomycin, and teicoplanin. (**D**) Evidence chart of clinical cure rate of MRSA-induced cSSSIs of seven treatment methods: minocycline and rifampin, linezolid, vancomycin, daptomycin, cefadroxi, cefazolin, and vancomycin and aztreonam. (**E**) Evidence chart of clinical cure rate of MRSA-induced cSSTIs of five treatment methods: tedizolid, linezolid, telavancin, vancomycin, and arbekacin. (**F**) Evidence chart of clinical cure rate of patients with MRSA-induced pneumonia of four treatment methods: vancomycin and rifampin, cefadroxil, linezolid, and vancomycin.

**Figure 3 antibiotics-13-00866-f003:**
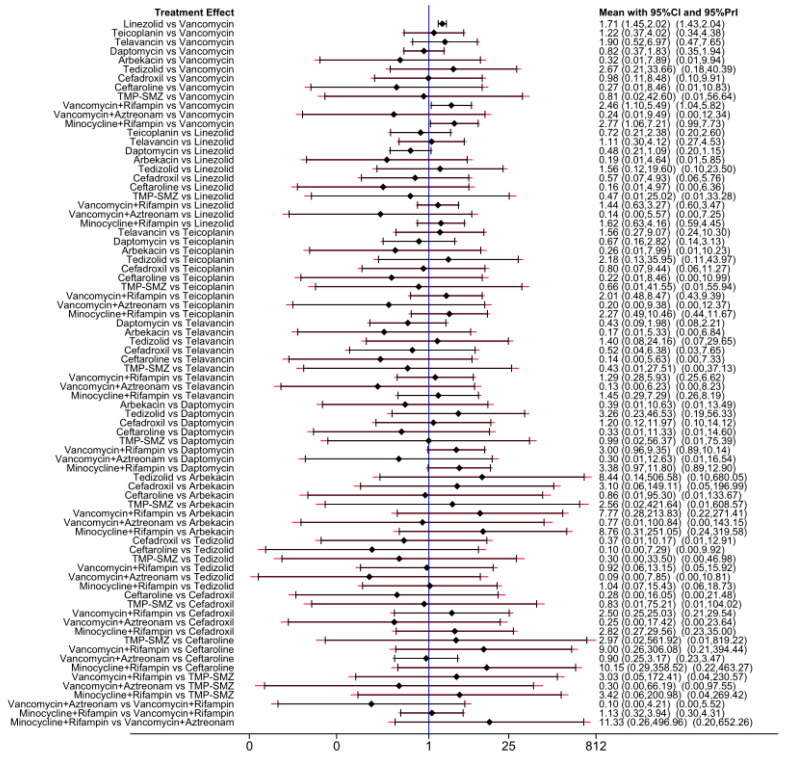
Forest plot of the network meta-analysis for clinical cure success. Includes 13 treatments: minocycline and rifampin, vancomycin and rifampin, tedizolid, telavancin, linezolid, teicoplanin, cefadroxil, TMP-SMZ, vancomycin, daptomycin, arbekacin, ceftaroline, and vancomycin and aztreonam.

**Figure 4 antibiotics-13-00866-f004:**
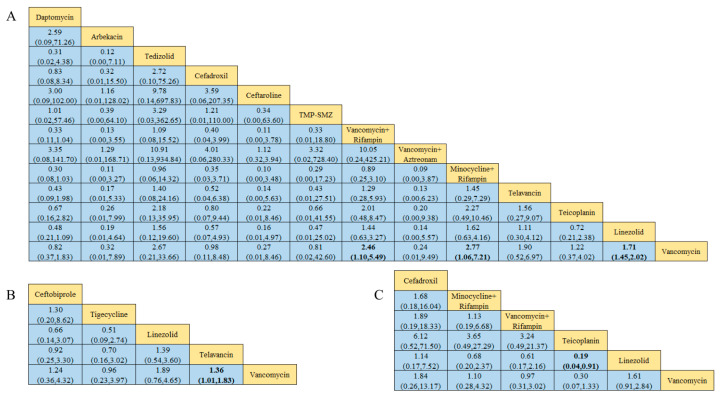
Results of the network meta-analysis, with bold numbers indicating significant differences: (**A**) Clinical cure success rate. (**B**) Clinical microbiology success rate. (**C**) Incidence of adverse reactions.

**Figure 5 antibiotics-13-00866-f005:**
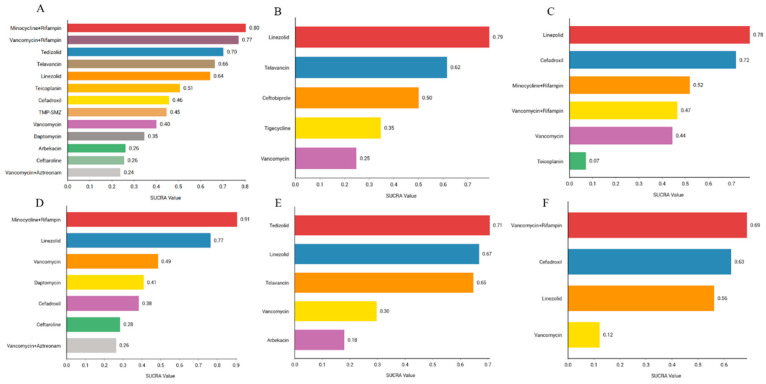
Efficacy and safety rankings of drugs used to treat MRSA infections: (**A**) Ranking of clinical cure rates for 13 treatments: minocycline and rifampin, vancomycin and rifampin, tedizolid, telavancin, linezolid, teicoplanin, cefadroxil, TMP-SMZ, vancomycin, daptomycin, arbekacin, ceftaroline, and vancomycin and aztreonam. (**B**) Ranking of clinical microbiology success rates for five treatments: linezolid, telavancin, ceftobiprole, tigecycline, and vancomycin. (**C**) Ranking of incidence of adverse reactions for six treatments: linezolid, cefadroxil, minocycline and rifampin, vancomycin and rifampin, vancomycin, and teicoplanin. (**D**) Ranking of clinical cure rate of MRSA-induced cSSSIs for seven treatments: minocycline and rifampin, linezolid, vancomycin, daptomycin, cefadroxi, cefazolin, and vancomycin and aztreonam. (**E**) Ranking of clinical cure rate of MRSA-induced cSSTIs for five treatments: tedizolid, linezolid, telavancin, vancomycin, and arbekacin. (**F**) Ranking of clinical cure rate of patients with MRSA-induced pneumonia for four treatments: vancomycin and rifampin, cefadroxil, linezolid, and vancomycin.

**Figure 6 antibiotics-13-00866-f006:**
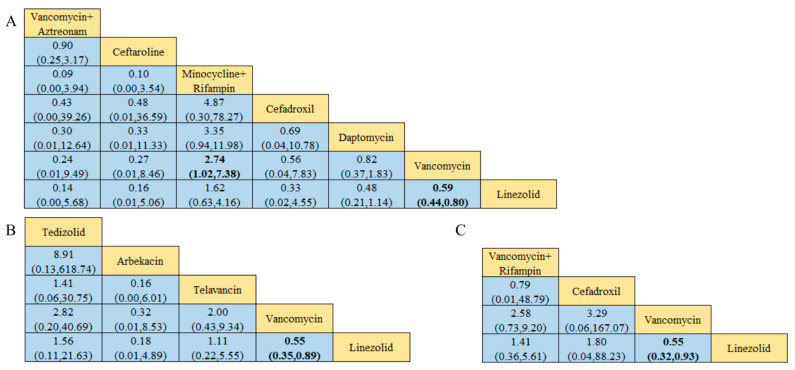
Graph of the results of the network meta-analysis, with bold numbers showing significant differences: (**A**) Clinical cure rate of MRSA-induced cSSSIs. (**B**) Clinical cure rate of MRSA-induced cSSTIs. (**C**) Clinical cure rate of patients with MRSA-induced pneumonia.

**Table 1 antibiotics-13-00866-t001:** Characteristics of the included studies (RCT: randomized controlled trial. AE: adverse event. CS: clinical success. ME: microbiologically evaluable. NP: nosocomial pneumonia.).

Study	Country	Age	Study Design	Types of MRSA Infection (Percentage of Infections)	Interventions	Sample Size	Dosage	Treatment Duration (Days)	Outcomes
Laethem 1988 [14]	Belgium	≥18	RCT, open-label	Infection (0.33%)	Vancomycin	9	i.v. 1 g qd	8–48	CS, AE
					Teicoplanin	12	i.v. or i.m. 400 mg qd		
Levine 1991 [15]	America	23–61	RCT	Endocarditis (0.67%)	Vancomycin	22	i.v. 1 g q12h	28	CS, AE
					Vancomycin+rifampin	20	i.v. or p.o. 1 g q12h and 400 mg qd		
Markowitz 1992 [16]	America	≥18	RCT	Infection (0.75%)	TMP-SMZ	21	i.v. 320 mg TMP and 1600 mg SMZ q12h		CS
					Vancomycin	26	i.v. 1g q12h		
Liu 1996 [17]	China	≥18	RCT	Bacteremia (0.64%)	Teicoplanin Vancomycin	20 20	i.v. 400 mg q12h for the first three times, then 400 mg qd		CS, AE
							i.v. 500 mg q6h		
Rubinstein 2001 [18]	Multinational	≥18	RCT, double-blind	NP (0.51%)	Linezolid + aztreonam	23	i.v. 600 mg q12h and 1–2g q8h	7–21	ME
					Vancomycin + aztreonam	9	i.v. 1 g q12h and 1–2 g q8h		
Stevens 2002 [19]	Multinational	≥13	RCT, open-label	SSTIs, pneumonia (2.31%)	Linezolid	76	i.v. 600 mg q12h	7–28	CS
					Vancomycin	69	i.v. 1g q12h		
Wunderink 2003 [20]	Multinational	≥18	RCT, double-blind	NP (1.96%)	Linezolid	61	i.v. 600 mg q12h	7–21	CS, AE
					Vancomycin	62	i.v. 1g q12h		
Kaplan 2003 [21]	America	5–17	RCT, open-label	cSSSIs, HAP (0.35%)	Linezolid	13	p.o. 10 mg/kg up to 600 mg/dose q12h	10–21	CS, AE
					Cefadroxil	9	p.o. 15 mg/kg up to 500 mg/dose q12h		
Kollef 2004 [22]	Multinational	≥65	RCT, double-blind	VAP (1.11%)	Linezolid	37	i.v. 600 mg q12h	7–21	CS
					Vancomycin	33	i.v. 1g q12h		
Weigelt 2004 [23]	Multinational	≥18	RCT, open-label	SSIs (1.70%)	Linezolid	53	i.v. or p.o. 600 mg q12h	7–21	CS
					Vancomycin	54	i.v. 1g q12h		
Sharpe 2005 [24]		≥18	Open-label	cSSTIs (0.96%)	Linezolid	30	p.o. 600 mg q12h	7–21	CS
					Vancomycin	30	i.v. 1g q12h		
Stryjewski 2005 [25]		≥18	RCT, double-blind	SSTIs (0.76%)	Telavancin	22	i.v. 600 mg q12h	>4	CS
					Vancomycin	26	i.v. 1g q12h		
Weigelt 2005 [26]	Multinational		RCT, open-label	SSTIs (5.75%)	Linezolid	176	i.v. or p.o. 600 mg q12h	7–21	CS
					Vancomycin	185	i.v. 1g q12h		
Stryjewski 2006 [27]	America, South Africa	≥18	RCT, double-blind	cSSSIs (0.72%)	Telavancin	26	i.v. 10mg/kg qd	4–14	ME
					Vancomycin	19	i.v. 1g q12h		
Talbot 2007 [28]	Multinational	≥18	RCT, single-blind	cSSSIs (0.16%)	Ceftaroline	5	i.v. 600 mg q12h	7–14	CS
					Vancomycin	5	i.v. 1 g q12h		
Kohno 2007 [29]	Japan	>20	RCT, open-label	cSSTIs, pneumonia, sepsis (1.46%)	Linezolid	62	i.v. or p.o. 600 mg q12h	7–28	ME
					Vancomycin	30	i.v. 1 g q12h		
Katz 2008 [30]	America	≥18	RCT	cSSSIs (0.94%)	Daptomycin	31	i.v. 10mg/kg qd	14	CS
					Vancomycin	28	i.v. 1 g q12h		
Florescu 2008 [31]	Multinational	≥18	RCT, double-blind	cSSSIs (1.31%)	Tigecycline	59	i.v. 100 mg q12h for the first four times, then 50 mg q12h	7–28	ME
					Vancomycin	23	i.v. 1 g q12h		
Noel 2008 [32]	Multinational	≥18	RCT, double-blind	cSSSIs (1.93%)	Ceftobiprole	61	i.v. 500 mg q12h		ME
					Vancomycin	60	i.v. 1 g q12h		
Stryjewski 2008 [33]	Multinational	≥18	RCT, double-blind	cSSSIs (9.22%)	Telavancin	278	i.v. 10mg/kg qd	7–14	ME
					Vancomycin	301	i.v. 1 g q12h		
Wilcox 2009 [34]	Multinational	≥13	RCT, open-label	cSSSIs (1.34%)	Linezolid	45	i.v. 600 mg q12h	7–28	CS
					Vancomycin	39	i.v. 1 g q12h		
Itani 2010 [35]	Multinational	≥18	RCT, open-label	cSSTIs (6.94%)	Linezolid	227	i.v. or p.o. 600 mg q12h	7–14	CS
					Vancomycin	209	i.v. 15 mg/kg q12h		
Jung 2010 [36]	Korea	≥18	RCT, open-label	Pneumonia (1.32%)	Vancomycin + rifampicin	41	i.v. 1 g q12h and 300 mg q12h	14	CS, AE
					Vancomycin	42	i.v. 1 g q12h		
Lipsky 2010 [37]	Multinational	Unclear	RCT, open-label	cSSSIs (11.26%)	Linezolid	356	i.v. or p.o. 600 mg q12h	7–28	CS, AE
					Vancomycin	351	i.v. 1 g q12h		
Corey 2010 [38]	Multinational	≥18	RCT, double-blind	cSSSIs (1.74%)	Ceftaroline	66	600 mg q12h	5–14	CS
					Vancomycin + aztreonam	43	1 g q12h and 1 g q12h		
Wilcox 2010 [39]	Multinational	≥18	RCT, double-blind	cSSSIs (1.75%)	Ceftaroline	58	600 mg q12h	5–14	CS
					Vancomycin + aztreonam	52	1 g q12h and 1 g q12h		
Corey 2010 [40]	Multinational	≥18	RCT, double-blind	cSSSIs (4.36%)	Ceftaroline	152	i.v. 600 mg q12h	5–14	ME
					Vancomycin + aztreonam	122	i.v. 1 g q12h and 1 g q12h		
Barriere 2010 [41]	Multinational	≥18	RCT, double-blind	cSSSIs (7.98%)	Telavancin	239	i.v. 10mg/kg qd	7–14	ME
					Vancomycin	262	i.v. 1 g q12h		
Duane 2012 [42]	Multinational	≥18	RCT, open-label	cSSSIs (2.52%)	Linezolid	73	i.v. 600 mg q12h	7–28	CS, AE
					Vancomycin	85	i.v. 1 g q12h		
Itani 2012 [43]	Multinational	≥18	RCT, open-label	cSSTIs (2.91%)	Linezolid	91	p.o. 600 mg q12h	7–14	CS, AE
					Vancomycin	92	i.v. 15 mg/kg q12h		
Wunderink 2012 [44]	Multinational	≥18	RCT, double-blind	Pneumonia (5.40%)	Linezolid	165	i.v. 600 mg q12h	7–14	CS
					Vancomycin	174	i.v. 15 mg/kg q12h		
Stryjewski 2012 [45]	Multinational	≥18	RCT, double-blind	cSSSIs (8.96%)	Telavancin	269	i.v. 10mg/kg qd	7–14	ME
					Vancomycin	294	i.v. 1 g q12h		
Stryjewski 2014 [46]	Multinational	≥18	RCT, double-blind	Bacteremia (0.14%)	Telavancin	5	i.v. 10mg/kg qd	14	CS
					Vancomycin	4	i.v. 1 g q12h		
Shaw 2015 [47]	America	≥18	RCT, open-label	cSSSIs (1.59%)	Daptomycin	50	i.v. 5 mg/kg qd	10–14	CS
					Vancomycin	50	i.v. 15 mg/kg q12h		
Equils 2016 [48]	Multinational	≥18	RCT, double-blind	NP (3.76%)	Linezolid	120	600 mg q12h	7–14	CS
					Vancomycin	116	15 mg/kg q12h		
Dube 2018 [49]	India	18–65	RCT, open-label	SSTIs (2.44%)	Arbekacin	75	200 mg injection OD	7–14	CS
					Vancomycin	78	1000 mg injection BD		
Mikamo 2018 [50]	Japan	≥18	RCT, open-label	SSTIs (0.57%)	Tedizolid	27	i.v. or p.o. 200 mg qd	7–21	CS
					Linezolid	9	i.v. or p.o. 600 mg q12h		
Kotsak 2023 [51]	Greece, Italy	≥18	RCT, open-label	cSSSIs (1.50%)	Minocycline + rifampicin	59 35	p.o. 600 mg qd and 100 mg q12h	10	CS, AE
					Linezolid		p.o. 600 mg q12h		

**Table 2 antibiotics-13-00866-t002:** Corresponding outcome indicators under different interventions and included references.

Interventions	Types of MRSA Infection	Outcomes	References
Vancomycin	cSSSIs, pneumonia, cSSTIs	CS, AE, ME	[14,15,16,17,19,20,22,23,24,25,26,27,28,29,30,31,32,33,34,35,36,37,41,42,43,44,45,46,47,48,49]
Linezolid	pneumonia, cSSSIs, cSSTIs	CS, AE, ME	[19,20,21,22,23,24,26,29,34,35,37,42,43,44,48,50,51]
Telavancin	cSSSIs	CS, ME	[25,27,33,41,45,46]
Ceftaroline	cSSSIs	CS, ME	[32,39,40]
Cefadroxil	cSSSIs, pneumonia	CS, AE	[21,28,38]
Daptomycin	cSSSIs	CS	[30,47]
Teicoplanin		CS, AE	[14,17]
Tedizolid	cSSTIs	CS	[50]
Arbekacin	cSSTIs	CS	[49]
Tigecycline	cSSSIs	ME	[31]
TMP-SMZ		CS	[16]
Minocycline + Rifampin	cSSSIs	CS, AE	[51]
Vancomycin + Rifampin	Pneumonia	CS, AE	[15,36]
Vancomycin + Aztreonam	Pneumonia, cSSSIs	CS, ME	[18,38,39,40]

## Data Availability

Not applicable.

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
