# Peer review of "Efficacy and Safety of Antibiotics in the Treatment of Methicillin-Resistant Staphylococcus aureus (MRSA) Infections: A Systematic Review and Network Meta-Analysis"

_antibiotics, 2024, doi:10.3390/antibiotics13090866_

Round 1

Reviewer 1 Report

Comments and Suggestions for Authors

Liu et al. performed a network meta-analysis to assess the (1) clinical and (2) microbiologic cure efficacy, and (3) the rate of adverse effects of alternative antibiotics to vancomycin in the (single or combined) treatment of methicillin-resistant Staphylococcus aureus (MRSA) infections. The analysis is mainly focused on subsets of data retrieved from 38 articles using different models to infer their conclusions, which is a relatively small set of data. Also, similar studies have been performed by other authors not so long ago. The authors seem to have used the appropriate research conduct and models to interpret the data and have, wisely, only made relative comparisons to rank the treatments. However, the paper has some major issues which need a major revision before being considered for acceptance.

Major issues:

-        The English of the article needs serious revision and hampers the full assessment of the data and the author claims. The grammar is often wrong, and the structure of the sentences are not correct. Also, the structure of the paper needs to be revised. The Tables and even the Figures need to be carefully designed for a better understanding. In some instances, it is very difficult to quickly grasp the results and understand the author claims. For instance, Table 1 describing the characteristics of the included studies is messy and difficult to read. The words on Figures 2, 3 and 6 need to be increased and, when necessary, the number of antibiotics analysed and the goal of the analysis (i.e.: clinical cure, microbiologic cure or rate of adverse effects) should be written as a subtitle of the different panels.

-        For each goal different subsets of the 38 articles are used. This is stated in each section, but the antibiotics analysed should be early stated in the text. To avoid repetition of the text, I would recommend the addition of a Table describing which antibiotics were analysed for each outcome and the associated references.

-        Please extend your description of the used methods in the Methods section. All your conclusions are based on this analysis, but the readers are completely unaware of the parameters and scripts used to perform the random forest analysis for the direct meta-analysis. The same is true for the SUCRA analysis and the funnel plots. Please add some information about your analysis so it can be reproducible by other researchers.

Other points:

-        Please avoid wrong terminology. I understand avoiding unnecessary jargon, but it is incorrect to call “investigators” or “researchers” to patients of the studies. Also, it is wise to avoid the use of informal words such as “kind” to describe different “class” of antibiotics. Finally, avoid using abbreviations without defining them first (i.e.: cSSSIs, cSSTIs on the Abstract, HAP/VAP, etc).

Comments on the Quality of English Language

English requires major revision. At this point it even hampers the correct assessment of the analysis performed. The study seems to have performed correctly and the conclusions are interesting. I look forward to read an improved version of the manuscript.

Author Response

Comments 1: The English of the article needs serious revision and hampers the full assessment of the data and the author claims. The grammar is often wrong, and the structure of the sentences are not correct. Also, the structure of the paper needs to be revised. The Tables and even the Figures need to be carefully designed for a better understanding. In some instances, it is very difficult to quickly grasp the results and understand the author claims. For instance, Table 1 describing the characteristics of the included studies is messy and difficult to read. The words on Figures 2, 3 and 6 need to be increased and, when necessary, the number of antibiotics analysed and the goal of the analysis (i.e.: clinical cure, microbiologic cure or rate of adverse effects) should be written as a subtitle of the different panels.

Response 1: Thank you for your questions and comments. We have carefully revised the English of the article through polishing, and corrected the grammar and sentence structure. First, we have modified Table 1, which describes the features included in the study, to make it more intuitive and convenient for readers. The subheadings of Figures 2, 3 and 6 were supplemented by adding antibiotic data and analysis targets.

Lines 184-197:

“Figure 2. Network evidence plot, where the size of nodes corresponds to the cumulative sample size for individual antibiotics, and the thickness of the lines is proportional to the number of studies for each treatment comparison: (A) Evidence chart of clinical cure rate of 13 treatment methods: minocycline and rifampin, vancomycin and rifampin, tedizolid, telavancin, linezolid, teicoplanin, cefadroxil, TMP-SMZ, vancomycin, daptomycin, arbekacin, ceftaroline, vancomycin and aztreonam. (B) Evidence chart of clinical microbiological success rate of 5 treatment methods: linezolid, telavancin, ceftobiprole, tigecycline, vancomycin. (C) Evidence chart of incidence of adverse effects of 6 treatment methods: linezolid, cefadroxil, minocycline and rifampin, vancomycin and rifampin, vancomycin, teicoplanin. (D) Evidence chart of clinical cure rate of MRSA-induced cSSSIs of 7 treatment methods: minocycline and rifampin, linezolid, vancomycin, daptomycin, cefadroxi, cefazolin, vancomycin and aztreonam. (E) Evidence chart of clinical cure rate of MRSA-induced cSSTIs of 5 treatment methods: tedizolid, linezolid, telavancin, vancomycin, arbekacin. (F) Evidence chart of clinical cure rate of patients with MRSA-induced pneumonia of 4 treatment methods: vancomycin and rifampin, cefadroxil, linezolid, vancomycin.”

Lines 200-203:

“Figure 3. Forest plot of the network meta-analysis for clinical cure success. Includes 13 treatments: minocycline and rifampin, vancomycin and rifampin, tedizolid, telavancin, linezolid, teicoplanin, cefadroxil, TMP-SMZ, vancomycin, daptomycin, arbekacin, ceftaroline, vancomycin and aztre-onam.”

Lines 291-302:

“Figure 6. Efficacy and safety rankings of drugs used to treat MRSA infections: (A) Ranking of clinical cure rates for 13 treatments: minocycline and rifampin, vancomycin and rifampin, tedizolid, telavancin, linezolid, teicoplanin, cefadroxil, TMP-SMZ, vancomycin, daptomycin, arbekacin, ceftaroline, vancomycin and aztreonam. (B) Ranking of clinical microbiology success rates for 5 treatments: linezolid, telavancin, ceftobiprole, tigecycline, vancomycin. (C) Ranking of incidence of adverse reactions for 6 treatments: linezolid, cefadroxil, minocycline and rifampin, vancomycin and ri-fampin, vancomycin, teicoplanin. (D) Ranking of clinical cure rate of MRSA-induced cSSSIs for 7 treatments: minocycline and rifampin, linezolid, vancomycin, daptomycin, cefadroxi, cefazolin, vancomycin and aztreonam. (E) Ranking of clinical cure rate of MRSA-induced cSSTIs for 5 treatments: tedizolid, linezolid, telavancin, vancomycin, arbekacin. (F) Ranking of clinical cure rate of patients with MRSA-induced pneumonia for 4 treatments: vancomycin and rifampin, cefadroxil, linezolid, vancomycin.”

Comments 2: For each goal different subsets of the 38 articles are used. This is stated in each section, but the antibiotics analysed should be early stated in the text. To avoid repetition of the text, I would recommend the addition of a Table describing which antibiotics were analysed for each outcome and the associated references.

Response 2: Thank you for your comments and suggestions. Based on your suggestions, we have added a new table 2 to show the outcome indicators corresponding to different interventions and the included literature.

Lines 179-180:

Table 2 Corresponding outcome indicators under different interventions and included references.

Interventions

Types of MRSA Infection

Outcomes

References

Vancomycin

cSSSIs, pneumonia, cSSTIs

CS, AE,ME

[14-17, 19, 20, 22-37, 41-46, 48, 49, 52]

Linezolid

pneumonia, cSSSIs, cSSTIs

CS, AE, ME

[19-24, 26, 29, 34, 35, 37, 42-44, 48, 50, 51]

Telavancin

cSSSIs

CS, ME

[25, 27, 33, 41, 45, 46]

Ceftaroline

cSSSIs

CS, ME

[32, 39, 40]

Cefadroxil

cSSSIs, pneumonia

CS, AE

[21, 28, 38]

Daptomycin

cSSSIs

CS

[30, 52]

Teicoplanin

CS, AE

[14, 17]

Tedizolid

cSSTIs

CS

[50]

Arbekacin

cSSTIs

CS

[49]

Tigecycline

cSSSIs

ME

[31]

TMP-SMZ

CS

[16]

Minocycline+Rifampin

cSSSIs

CS, AE

[51]

Vancomycin+Rifampin

Pneumonia

CS, AE

[15, 36]

Vancomycin+Aztreonam

Pneumonia,cSSSIs

CS, ME

[18, 38-40]

Comments 3: Please extend your description of the used methods in the Methods section. All your conclusions are based on this analysis, but the readers are completely unaware of the parameters and scripts used to perform the random forest analysis for the direct meta-analysis. The same is true for the SUCRA analysis and the funnel plots. Please add some information about your analysis so it can be reproducible by other researchers.

Response 3: Thank you for your comments and suggestions. We have made a detailed description in the description of data statistics in the method part, and have also made supplements and corrections in response to your questions.

Lines 122-138:

“We evaluated three outcome measures as well as subgroup analyses of three diseases in clinical success measures through a network meta-analysis, comparing the effectiveness and safety of various antibiotics for the treatment of MRSA infections. A meta-analysis of bicategorical variables was performed using the meta software pack-age in R language. Relative risks (RRs) and 95% confidence intervals (95%-CIs) were calculated to compare the efficacy and safety of various antibiotics in a specified population. If 95%-CI of RR does not contain 1, it is considered that there is a statistically significant difference. The size of the heterogeneity was assessed using I2. If I2 < 50%, the effect size was combined using a fixed effect model, and if I2≥50%, the effect size was combined using a random effect model. Using Stata 14.0 software, a network me-ta-analysis was conducted based on the random effects model under the frequency framework, and the network evidence map was drawn. Subsequently, the cumulative ranking probability area under graph (SUCRA) was calculated by the “sucra prob” command to predict the efficacy ranking of each intervention. The larger the SUCRA value, the better the effectiveness of the intervention, so as to facilitate the direct and indirect comparison of the efficacy of various antibiotics. Finally, the “netfunnel” command was used to plot funnel plots to evaluate publication bias in the included literature. This study’s PROSPERO registration number is CRD42023416788.”

Other points:

Comments 4: Please avoid wrong terminology. I understand avoiding unnecessary jargon, but it is incorrect to call “investigators” or “researchers” to patients of the studies. Also, it is wise to avoid the use of informal words such as “kind” to describe different “class” of antibiotics. Finally, avoid using abbreviations without defining them first (i.e.: cSSSIs, cSSTIs on the Abstract, HAP/VAP, etc).

Response 4: Thank you for your comments and suggestions. We have avoided some unnecessary terms and added abbreviations that were not defined in the article.

Lines 18-22:

“Methods: All studies were obtained from the PubMed and Embase databases from inception to April 13, 2023. The three comprehensive indicators of clinical cure success rate, clinical microbiological success rate, and adverse reactions were evaluated, and the clinical cure success rates of three disease types, complex skin and skin structure infections (cSSSIs), complex skin and soft tissue infections (cSSTIs), and pneumonia, were analyzed in subgroups.”

Lines 325-326:

“Three similar studies were conducted previously. In 2012, Bally et al. compared the efficacy of six antibiotics used to treat cSSTIs and HAP/VAP(Hospital-Acquired Pneumonia/ Ventilator-associated Pneumonia),”

Reviewer 2 Report

Comments and Suggestions for Authors

This study conducted a network meta-analysis (NMA) of studies sourced from the literature (MeSH) to compare the efficacy and safety of different antibiotics against vancomycin, the most commonly utilized antibiotic used in the treatment of methicillin resistant Staphylococcus aureus (MRSA)-involved infections.  The goal was to determine which antibiotics were the best alternative to vancomycin, since vancomycin resistance is becoming more prevalent, making treatment of MRSA infections difficult.  It is a good study that has minor critiques of the manuscript that can be improved with adding more information to inform the reader not familiar with some of the methods.

This study depends heavily on the methods used to analyze the data in the published, peer-reviewed articles.  The descriptions of the methodology could be improved, especially in the retrieval and the analysis of the data. Data retrieval was from articles as late as 2023. What year was the earliest study that was analyzed? Was antibiotic sensitivity testing done in these studies? How was MRSA identified? How was the test of cure endpoint specifically defined?  What were the adverse events observed?

More description is needed to elucidate the programs used to generate risk analysis, such as Review Manager to generate the Risk of Bias map. A description of the random effects model to obtain relative risks and 95% confidence intervals is needed. It is necessary to include a better description of how R and Stata are used to analyze the data.

Author Response

Comments 1: This study depends heavily on the methods used to analyze the data in the published, peer-reviewed articles.  The descriptions of the methodology could be improved, especially in the retrieval and the analysis of the data. Data retrieval was from articles as late as 2023. What year was the earliest study that was analyzed? Was antibiotic sensitivity testing done in these studies? How was MRSA identified? How was the test of cure endpoint specifically defined?  What were the adverse events observed?

Response 1: Thank you for your questions and suggestions, we will answer your questions one by one. First, we searched the data from the inception of the database until May 2023, with the earliest analysis in 1991. Second, some of the studies we included conducted clinical trials while conducting antibiotic sensitivity tests, while articles that only conducted antibiotic sensitivity tests were not included in this paper. Third, as to how MRSA was determined, we found through reading the included literature that the identification of MRSA was confirmed by bacterial Gram staining, culture and drug susceptibility tests. Fourth, test of cure (TOC) is defined as an assessment performed 7-14 days after the end of treatment. Finally, we added the observed adverse reactions, including nausea, diarrhea, headache, and thrombocytopenia.

Lines 75-77:

“We systematically searched PubMed and EMBASE for potentially eligible studies (database was established until April 2023) using Medical Subject Headings (MeSH) descriptors combined with free-text terms to retrieve two categories:”

Lines 82-89:

“This study included articles that complied with the following criteria: (1) RCTs or observational studies published in English. (2) Studies including patients diagnosed with MRSA infection (The diagnosis was confirmed by Gram staining, culture and drug sensitivity test), as well as patients with cSSSIs, cSSTIs, and pneumonia caused by MRSA infection, comparing antibiotics with anti-MRSA activity against one another. (3) The primary outcomes were the rate of clinical and microbiologic cure at the test-of-cure visit and the rate of adverse effects. In addition, we excluded relevant literature in which only antibiotic experiments were conducted.”

Lines 104-111:

“Clinical success was defined as cure or improvement to assess the status of the study population upon the test of cure (TOC), TOC is assessed 7-14 days after the end of treatment. For microbiological success, we assessed the microbiologically evaluable (ME) population as patients who had at least one MRSA pathogen isolated from blood or infected tissue at baseline in the CE population, with microbiological success defined as the eradication of MRSA pathogens. Finally, safety, defined as the incidence of ad-verse events, was assessed. Some of the common adverse reactions include nausea, diarrhea, headache, and thrombocytopenia.”

Comments 2: More description is needed to elucidate the programs used to generate risk analysis, such as Review Manager to generate the Risk of Bias map. A description of the random effects model to obtain relative risks and 95% confidence intervals is needed. It is necessary to include a better description of how R and Stata are used to analyze the data.

Response 2: Thank you for your questions and comments. In the methods section, we elaborated the factors of risk assessment in more detail. We plotted the risk shift map by evaluating the risk degree of random sequence generation, allocation hiding, participant and personnel blindness, outcome evaluation blindness, outcome data incompleteness, selective reporting, and other seven indicators of bias. The random effects model and how to use R and STATA are described.

Lines 122-138:

“We evaluated three outcome measures as well as subgroup analyses of three diseases in clinical success measures through a network meta-analysis, comparing the effectiveness and safety of various antibiotics for the treatment of MRSA infections. A meta-analysis of bicategorical variables was performed using the meta software pack-age in R language. Relative risks (RRs) and 95% confidence intervals (95%-CIs) were calculated to compare the efficacy and safety of various antibiotics in a specified population. If 95%-CI of RR does not contain 1, it is considered that there is a statistically significant difference. The size of the heterogeneity was assessed using I2. If I2 < 50%, the effect size was combined using a fixed effect model, and if I2≥50%, the effect size was combined using a random effect model. Using Stata 14.0 software, a network meta-analysis was conducted based on the random effects model under the frequency framework, and the network evidence map was drawn. Subsequently, the cumulative ranking probability area under graph (SUCRA) was calculated by the “sucra prob” command to predict the efficacy ranking of each intervention. The larger the SUCRA value, the better the effectiveness of the intervention, so as to facilitate the direct and indirect comparison of the efficacy of various antibiotics. Finally, the “netfunnel” command was used to plot funnel plots to evaluate publication bias in the included literature. This study’s PROSPERO registration number is CRD42023416788."

Round 2

Reviewer 1 Report

Comments and Suggestions for Authors The authors have addressed most of my previous comments and the manuscript has improved greatly. In particular, the English has been dramatically improved and only minor corrections are required here and there. Still, I am not yet satisfied with Figure 2, where the name of the antibiotics are too small to be read. Comments on the Quality of English Language The English has dramatically improved. Still some minor typos here and there but the message is clear.